# Integrated Analysis of the Intestinal Microbiota and Transcriptome of *Fenneropenaeus chinensis* Response to Low-Salinity Stress

**DOI:** 10.3390/biology12121502

**Published:** 2023-12-07

**Authors:** Caijuan Tian, Qiong Wang, Jiajia Wang, Jitao Li, Chenhui Guan, Yuying He, Huan Gao

**Affiliations:** 1Jiangsu Key Laboratory of Marine Bioresources and Environment/Jiangsu Key Laboratory of Marine Biotechnology, Jiangsu Ocean University, Lianyungang 222005, China; tcaijuan@163.com; 2National Key Laboratory of Mariculture Biobreeding and Sustainable Goods, Yellow Sea Fisheries Research Institute, Chinese Academy of Fishery Sciences, Qingdao 266071, China; wangqiong@ysfri.ac.cn (Q.W.); wangjj@ysfri.ac.cn (J.W.); lijt@ysfri.ac.cn (J.L.); guanchenhui678@163.com (C.G.); 3Function Laboratory for Marine Fisheries Science and Food Production Processes, Pilot National Laboratory for Marine Science and Technology, Qingdao 266200, China; 4School of Marine Science and Engineering, Qingdao Agricultural University, Qingdao 266237, China

**Keywords:** *Fenneropenaeus chinensis*, intestinal microbiota, transcriptome, immune response, low salinity

## Abstract

**Simple Summary:**

This study aimed to investigate the influencing mechanisms of the intestinal microbiota and gene expression of *Fenneropenaeus chinensis* under low-salinity stress. The 16S rDNA results suggest that the relative abundances of *Photobacterium* and *Vibrio* decreased significantly, whereas some bacteria, *Shewanella*, *Pseudomonas*, and *Lactobacillus*, became the predominant communities. Transcriptome sequencing identified numerous differentially expressed genes (DEGs) through various types of N-glycan biosynthesis, amino acid sugar and nucleotide sugar metabolism, and lysosomes to adapt to stress. Correlation analysis between microbiota and DEGs showed that significant changes in *Pseudomonas*, *Ralstonia*, *Colwellia*, and *Cohaesibacter* were positively correlated with immune-related genes such as peritrophin-1-like and mucin-2-like, and negatively correlated with caspase-1-like.

**Abstract:**

Salinity is an important environmental stress factor in mariculture. Shrimp intestines harbor dense and diverse microbial communities that maintain host health and anti-pathogen capabilities under salinity stress. In this study, 16s amplicon and transcriptome sequencing were used to analyze the intestine of *Fenneropenaeus chinensis* under low-salinity stress (15 ppt). This study aimed to investigate the response mechanisms of the intestinal microbiota and gene expression to acute low-salinity stress. The intestinal tissues of *F. chinensis* were analyzed using 16S microbiota and transcriptome sequencing. The microbiota analysis demonstrated that the relative abundances of *Photobacterium* and *Vibrio* decreased significantly, whereas *Shewanella*, *Pseudomonas*, *Lactobacillus*, *Ralstonia*, *Colwellia*, *Cohaesibacter*, *Fusibacter*, and *Lachnospiraceae_NK4A136_group* became the predominant communities. Transcriptome sequencing identified numerous differentially expressed genes (DEGs). The DEGs were clustered into many Gene Ontology terms and further enriched in some immunity- or metabolism-related Kyoto Encyclopedia of Genes and Genomes pathways, including various types of N-glycan biosynthesis, amino acid sugar and nucleotide sugar metabolism, and lysosome and fatty acid metabolism. Correlation analysis between microbiota and DEGs showed that changes in *Pseudomonas*, *Ralstonia*, *Colwellia*, and *Cohaesibacter* were positively correlated with immune-related genes such as peritrophin-1-like and mucin-2-like, and negatively correlated with caspase-1-like genes. Low-salinity stress caused changes in intestinal microorganisms and their gene expression, with a close correlation between them.

## 1. Introduction

*Fenneropenaeus chinensis*, a prominent aquatic species in the Yellow and Bohai Seas of China and the Peninsula coast of Korea [1], has considerable economic and nutritional value and a delectable flavor [2,3]. Its production in China has increased substantially since 2003 [4] and can be adjusted by varying environmental salinities from 18 to 32 [5]. Shrimp culture has demonstrated a tendency to move increasingly from coastal to inland areas, making great use of natural water bodies and minimizing the impact on coastal ecological areas [6,7], owing to the ongoing development of desalination culture technology. Further, low-salinity water is frequently used in several countries, including China, the United States, Thailand, Brazil, and Mexico, to produce inland shrimps [8,9,10].

Environmental stress, including dissolved oxygen [11,12], temperature [13,14], salinity [14,15], and ammonia nitrogen stresses [13], generally serves as an adaptation mechanism to survive and benefit from potentially threatening environments. Appropriate salinity modulates the microbiota composition and the immune response of host animals in a presumably beneficial manner [16]. In marine species, low-salinity stress typically leads to oxidative damage, physiological malfunction, and immunological disease [17,18,19]. Generally, shrimps maintain a balance of osmotic pressure through the hemolymph system to ensure immune homeostasis in the internal environment [20]. 

Several studies have shown that the composition of intestinal microbiota of aquatic animals is mainly determined by biotic factors, such as host developmental stages [21,22] and dietary habits [23,24], whereas abiotic factors have been less studied. The digestive tracts of animals with sophisticated microecosystems are home to a sizable microbial community of between 1000 and 5000 species. The animal intestine acts as a portal for pathogenic colonies and invades host nutrient absorption [25], metabolism [26], immunity [27,28,29], and osmoregulation [30] homeostasis. Dysbiosis of the intestinal microbiota may disrupt the host intestinal barrier and lead to pathogenic infections and diseases by reducing the ability of the microbiota to guide the development of the immune system [31]. The immune mechanisms in the intestines of crustaceans have recently become an important research subject. Therefore, it is critical to elucidate how the microbiota affect the development, growth, and cellular functions of host organisms under low salinity. This study aims to provide key insights into the following: (1) the influencing mechanism of intestinal microbiota and differential expression of some genes subjected to salinity stress; and (2) the correlation between changes in intestinal microbiota and immune gene expression.

## 2. Materials and Methods

### 2.1. Shrimp Rearing and Sample Collection

Healthy *F. chinensis* shrimps (body weight = 9.53 ± 1.55 g) were obtained from the Haifeng Fishery Technology Company (Changyi, China). The included male and female shrimps were about 4 months old. A total of 150 shrimps were acclimated in aerated seawater (salinity 30.02 ± 0.44 ppt, pH 8.47 ± 0.17, dissolved oxygen 5.02 ± 0.03 mg/L, temperature 27.9 ± 1.9 °C) for 7 days. These water-quality parameters were detected three times and the average value was taken using a Multi-parameter Water Quality Analyzer (YSI, EXO2, Yellow Springs, OH, USA). One third of the tank volume of seawater was changed daily. Oxygenation pumps were continuously operated, and *F. chinensis* were fed with fresh wild fish and shrimps at 8:00 a.m. and 6:00 p.m. This experiment included a low-salinity stress group (salinity 15 ppt) and a control group [32] (salinity 30 ppt, C0), and 50 shrimps were cultured in each group and 9 shrimps (three parallels, three replicates) were used to collect samples in each group. The experimental conditions were consistent with those of the acclimatization period. During the experiment, salinity was corrected every 6 h using light brine. Intestinal tissues were collected after exposure to low-salinity stress on days 3 (S3), 7 (S7), and 14 (S14), extracted aseptically, and placed in 1.5 mL sterile centrifuge tubes for microbiome and transcriptome sequencing. All the samples were immediately frozen in liquid nitrogen and stored in a −80 °C refrigerator until analysis.

### 2.2. Microbiota Analysis Using 16S rDNA Genes Sequencing

The intestinal microbiota of the shrimps was examined using Illumina sequencing (NovaSeq, Illumina, San Diego, CA, USA) of the V4 region of the 16S rDNA genes. The whole genomic DNA of the sample was extracted using cetyltrimethylammonium bromide/sodium dodecyl sulfate methods and detected in 12 samples. The purity and concentration of the extracted DNA were detected using agarose gel electrophoresis. Using diluted genomic DNA as a template, a polymerase chain reaction (PCR) was performed using the following primers: 515 F (5′-GTGCCAGCMGCCGCGGTAA-3′ and 806 R (5′-GGACTACHVGGGTWTCTAAT-3′).

Libraries were constructed using the TruSeq^®^ DNA PCR Free Sample Preparation Kit (Illumina, San Diego, CA, USA), quantified with Qubit, and then sequenced with NovaSeq 6000 (TruSeq DNA PCR-Free Library Preparation Kit). The final Amplicon Sequence Variants (ASVs) and feature lists were obtained using the deblur module in DADA2 [33] to reduce noise and filter out sequences with an abundance of less than 5 species annotation using the Silva 138.1 database. Alpha and beta diversity indices were calculated using the QIIME2 software (Version QIIME2-202006, Caporaso Lab, Flagstaff, AZ, USA), and the rarefaction curves were plotted. Principal Co-ordinates Analysis (PcoA)were analyzed based on the UniFrac distance of weighted and unweighted [34]. If the samples are closer to each other, the species’ compositional structure is more similar. Linear discriminant analysis Effect Size (LEfSe) is an analytical tool for discovering and interpreting high-dimensional biomarkers, including histograms of the distribution of linear discriminant analysis (LDA) values and evolutionary branching maps [35]. Finally, the significantly different species between groups were analyzed using LEfSe, LDA score > 4.

### 2.3. Gene Expression Analysis Using RNA Sequencing

Total RNA was extracted from all intestinal tissues using TRIzol reagent (Thermo Fisher Scientific, Waltham, MA, USA) according to the manufacturer’s instructions. The RNA integrity was assessed using the RNA Nano 6000 Assay Kit for the Bioanalyzer 2100 System (Agilent Technologies, Santa Clara, CA, USA). The library construction kit was NEBNext^®^ Ultra™ RNA Library Prep Kit for Illumina^®^. All clean paired-end reads were separately aligned to the *F. chinensis* genome (ASM1920278v2) using HISAT2.0.5. We used DESeq2 to detect differentially expressed genes (DEGs), using fragments per kilobase per million mapped reads (FPKM) by StringTie (v1.3.3b, The Center for Computational Biology at Johns Hopkins University, Baltimore, MD, USA), which were defined as genes with |log2(FoldChange)| > 1 and *p*-value < 0.05 between the normal salinity and low-salinity stress groups. DEGs enrichment analyses of Gene Ontology (GO) and Kyoto Encyclopedia of Genes and Genomes (KEGG) were performed using ClusterProfiler (3.8.1) software (v2.0.5).

### 2.4. Verification of Real-Time Quantitative PCR (qPCR)

To verify the expression patterns of the genes identified by RNA-seq, quantitative reverse transcriptase (qRT)-PCR analyses of some selected genes were performed on a 7500 Fast Real-Time PCR system (Applied Biosystems, Foster, CA, USA) using the SYBR Green PCR Master Mix (Life Technologies, MASS, Waltham, MA, USA) with the same extracted RNA samples as those used for transcriptome sequencing. β-actin was used as an internal control to normalize the cycle threshold (CT) values of the target genes. The expression of the target genes was calculated using the 2^−ΔΔCt^ method [36], and all the reactions were repeated in triplicate to ensure the reproducibility of the results.

### 2.5. Correlations between ASVs and DEGs

Using Cytoscape software (Cytoscape 3.8.0, San Diego, CA, USA), Pearson correlation analysis was used to reveal the correlation between intestinal DEGs expression and intestinal bacteria in the host: values with *p* < 0.05 were considered statistically significant. Immune-related DEGs were filtered using |log2(foldchange)| > 1 and *p*-value < 0.01. Significant differences among groups were estimated by one-way analysis of variance followed by Duncan’s multiple range tests using the Statistical Package for the Social Sciences (SPSS 22.0, IBM, Armonk, NY, USA).

## 3. Results

### 3.1. Intestinal Microbiota Community Analysis

A total of 992,563 clean reads were obtained from the intestine of *F. chinensis* using 16S rDNA sequencing, with an average of 82,714 reads per sample obtained after optimization and quality control (Appendix A). The rarefaction of each sample indicated that the depth and quality of the sequencing were sufficient (Appendix A). For the intestines collected from shrimp treated with low salinity for 14 days, unique ASVs were shown, whereas 248 ASVs were shared by all four groups (Figure 1A,B). Alpha diversity analysis showed that the Chao 1, Shannon, and Simpson indices were decreased significantly (*p* < 0.05) in S3 under low-salinity conditions compared with C0, while the dominance index increased significantly (*p* < 0.05). It exhibited the opposite tendency in S7 vs. C0. In S14, these indices gradually returned to the level of normal shrimp (Figure 1C–F, Appendix A). Principal coordinate analysis, based on unweighted and weighted UniFrac distances, demonstrated the heterogeneity and diversity of the species composition of intestinal samples at various times (Appendix A).

Low-salinity stress altered the structure and composition of the intestinal microbiota of *F. chinensis*. At the phylum level (Figure 2A, Appendix A), the dominant phyla of the intestinal microbial community of shrimps were Proteobacteria, Firmicutes, and Actinobacteria, and Proteobacteria was the most abundant phylum, accounting for 38.06–70.53% of all groups. Compared with C0, the relative abundance of Proteobacteria increased significantly (*p* < 0.05) in the S3 group, while it was greatly reduced in S7 and remained stable in S14, just opposite to Actinobacteria, and was restored to normal levels. Firmicutes decreased markedly in S3, but increased significantly (*p* < 0.05) in S14. At the genus level, for all groups at C0, the dominant genera in the three groups showed different results. The genera (Figure 2B, Appendix A) *Photobacterium* and *Vibrio* were the primary intestinal bacteria in the C0. Compared to C0, *Photobacterium* and *Sphingomonas* abundances were enriched in S3. Subsequently, *Photobacterium* and *Vibrio* decreased dramatically in S7 and S14, respectively.

At the phylum and genus level, the top 10 bacteria significantly affected by low-salinity stress were used for further analysis (Student’s *t*-test, *p* < 0.05, Figure 2C,D). To investigate the different abundances of bacterial taxa associated with salinity exposure, an LDA of the effect size between C0 and S14 was performed (Figure 2E,F). In the cladogram, Photobacterium was enriched in C0. The relative abundances of these eight genera were significantly increased in S14, *Ralstonia*, *Colwellia*, *Lactobacillus*, *Shewanella*, *Lachnospiraceae_NK4A136_group*, *Fusibacter*, *Pseudomonas*, and *Cohaesibacter.*

### 3.2. Intestinal Transcriptome Analysis

A total of 514,166,164 high-quality clean reads (77.13 G) (Appendix A) were obtained after removing poly-N, adaptor, and low-quality sequences, which were subsequently mapped to the reference genome (ASM1920278v2). The mapping rates ranged from 85.24 to 91.12% (Appendix A). Sums of 1352 (297 upregulated and 1055 downregulated), 314 (227 upregulated and 87 downregulated), and 639 (354 upregulated and 285 downregulated) DEGs were identified in S3, S7 and S14, respectively (Figure 3A).

To verify the intestinal transcriptome results, 10 DEGs were selected and primers were designed for real time quantitative PCR (qPCR) (Appendix A). The results of the qRT-PCR analysis were consistent with those of the RNA-seq data, revealing the reliability and robustness of the transcriptome analysis (Figure 3B,D).

A large number of GO terms were significantly enriched in S3, S7, and S14 among the differentially expressed transcripts compared with C0 (*p* < 0.05) (Appendix A). “proteolysis”, “carbohydrate derivative biosynthetic process”, “extracellular region”, “peptidase activity”, and “peptidase activity, acting on L-amino acid peptides” represented many DEGs in the S3 (Figure 4A). In the S7 and S14 groups, the DEGs were mainly involved in “transmembrane transport”, “lipid metabolic process”, “chitin binding”, “sulfotransferase activity”, “transferase activity”, and “transferring sulfur-containing groups” (Figure 4B,C).

The KEGG pathways were enriched in S3, S7, and S14 (Appendix A). In S3, 119 DEGs (18 upregulated and 101 downregulated) were significantly associated with various biosynthesis and energy metabolism processes (Figure 4D). In S7, 19 (15 upregulated and 4 downregulated) DEGs significantly changed in relation to lysosomes, various types of N-glycan biosynthesis, and the Notch and Wnt signaling pathways (Figure 4E). At S14, 112 (41 upregulated and 71 downregulated) DEGs were significantly enriched in related pathways, including fatty acid metabolism, lysosome, glycolysis/gluconeogenesis, amino acid biosynthesis, fatty acid biosynthesis, and nitrogen metabolism (Figure 4F).

### 3.3. Correlation between Intestinal Microbiota and Immune-Related DEGs

To understand the interaction mechanism between the intestinal microbiota and the gene expression response to low-salinity stress in *F. chinensis*, we selected 12 immune-related DEGs in S3, S7, and S14 (Appendix A). Additionally, the 30 most dominant microbial genera under low-salinity conditions were selected to confirm the key microbial markers of salinity stress. 

The correlations between intestinal bacteria and the expression of immune genes were calculated (Figure 5). The changes in *Pseudomonas* and *Ralstonia* were significantly positively associated with peritrophin-1-like (*PT-1*), mucin-2-like (*Muc-2*), phospholipase A2 group XV-like (*PLA2G15*), procathepsin L-like (*pCTS-L*), and cathepsin B-like (*CTSB*), but were negatively correlated with caspase-1-like (*Casp1*). The genera of *Colwellia*, *Cohaesibacter*, and *Fusibacter* were significantly and positively correlated with *PT-1* expression and negatively correlated with *Casp1*. The *Lachnospiraceae_NK4A136_group* was significantly positively correlated with *PT-1*, *Muc-2*, tumor necrosis factor alpha-induced protein 8-like protein (*TNFAIP8*), *PLA2G15*, *pCTS-L*, and *CTSB*, and negatively correlated with *Casp1*. *Lactococcus* was significantly positively correlated with *Muc-2*, lysozyme (*Lys*), CD109 antigen-like (*CD109*), *TNFAIP8*, *pCTS-L*, and *CTSB* and negatively correlated with *Casp1*. *Photobacterium* was positively correlated with collagen alpha-5(IV) chain-like (*COL4α5*) and *Casp1* while being negatively correlated with *TNFAIP8*, *PLA2G15*, *pCTS-L*, and *CTSB*.

## 4. Discussion

### 4.1. Intestinal Microbial Community Changed under Low-Salinity Stress

Several studies have shown that environmental stress and pathogen challenges can alter the composition of the intestinal microbiota in aquatic animals [37,38,39]. In this novel study, we studied the effect of the intestinal microbiota of *F. chinensis* in response to low-salinity stress. High-throughput sequencing analysis of 16S rDNA revealed that low-salinity stress markedly disrupted the diversity and abundance in the intestine of shrimps in S3. This is because the shrimp were subjected to acute low-salinity stress, resulting in the disruption of the dynamic balance of the intestine. Shrimp were unable to satisfy the energy demand in a short period of time, and thus Alpha diversity was reduced in S3. With the prolongation of low-salinity stress, shrimp gradually adapted to various metabolic and energy levels in the body. Acute salinity stress may disrupt the intestinal microbiota balance, leading to an unhealthy state of the intestine and even death.

Proteobacteria, Firmicutes, Actinobacteria, and Bacteroidetes were the dominant phyla in the shrimp intestinal microbiota, consistent with previous studies [40,41]. Notably, we found that Proteobacteria, the most abundant phylum in the intestine of *F. chinensis*, imbalance is a potential marker of immune disorders in the intestine [42,43], thus enhancing the intestinal disorders in shrimp that typically occur under salinity stress. Additionally, the proportion of Bacteroidetes increased, whereas that of Firmicutes decreased in this study. The ratio of Firmicutes to Bacteroidetes can reflect the physiological status of the intestine, and a decrease in Firmicutes indicates an intestinal dysfunction [44,45,46]. Until S7, the abundances of Proteobacteria, Firmicutes, and Actinobacteria were gradually up to the level of normal shrimp compared to C0. This is because with the extension of stress, the shrimp slowly adapted to its physiological state. However, Firmicutes had the highest relative abundance in S14 under salinity stress, which is related to fermentation and provides nutrients to promote energy harvesting by the host [47] when shrimps are exposed to low-salinity stress. 

Although the genus *Photobacterium* [48], a marine pathogen, increased, the probiotic *Sphingomonas* [49] also significantly increased in the S3 group compared to that in the control group under low-salinity stress. The proportion of opportunistic bacteria increased, whereas those regarded as commensal or beneficial bacteria decreased when the host faced hyposaline stress. Particularly, the relative abundance of *Vibrio* [50], an opportunistic pathogenic bacterium, increases significantly when exposed to pathogenic infections and extreme environmental stress. However, its relative abundance decreased in this study, and possibly extreme environmental stress was not reached. In the S7 group, the relative abundance of the genera did not change significantly. Additionally, in S14, some potential probiotics, such as *Lactobacillus*, *Pseudomonas*, and *Shewanella*, were significantly increased, which may be conducive to balancing the intestinal microbiota and immune system [51,52,53], thus alleviating the simultaneous stress imposed by salinity. Based on the LEfSe analysis in this study, as low-salinity stress continued, bacteria associated with biodegradation, such as *Colwellia* [54], *Lachnospiraceae_NK4A136_group* [55] related to biological metabolism, and the pathogenic bacteria *Ralstonia* [56,57], also significantly increased. These bacteria may be responsible for enhancing intestinal immunity in the host to restore salinity stimulation.

### 4.2. Participation of Immune-Related Genes in Response to Low-Salinity Stress

As an important immune organ, the intestine maintains homeostasis under conditions of low-salinity stress by regulating the expression of immune-related genes [58]. Comprehensive profiling of the intestinal transcriptome will improve our understanding of the adaptability of shrimps to salinity. Due to the lack of adaptive immune mechanisms in crustaceans, researchers have paid increasing attention to the molecular mechanism of the innate immunity of shrimps, which is the first line of defense against invading pathogenic microorganisms [59,60]. Phagocytosis and apoptosis, which are critical in the cellular immune response, play significant roles in endogenous antiviral activity in shrimps [59,61]. In this study, we found that low salinity significantly affected the digestive and absorptive functions of the intestine and the expression of immune-related genes. For example, most genes significantly suppressed transcriptional expression in S3 under low-salinity stress, confirming that the shrimps were markedly susceptible to stress at this stage.

Mucin protein, an important component of the intestinal mucus layer, could protect the intestine from bacterial attachment and invasion, thereby inhibiting the local inflammatory reaction [62,63]. *Muc-2* and mucin-4-like (*Muc-4*) allow components of the microbiota to penetrate and reside within the mucus, affecting mucosal protection and increasing susceptibility to luminal insults [64,65]. However, the downregulation of *Muc-2* and *Muc-4* proteins showed that the intestine could be damaged in the deep layer of the mucosa in response to salinity stress. In contrast, collagen, the most abundant glycoprotein, is the main component of the extracellular matrix [66]. Here, we identified that *COL4α5* mRNA was significantly overexpressed on day 14 to alleviate the salinity-induced intestinal barrier damage, which indicates that low-salinity stress severely affected the intestinal cellular basement.

Organisms rely on antioxidant systems to generate oxidative stress from reactive oxygen species in response to pathogen attacks [67]. Several important immune antimicrobial peptides, *Lys* [68], peroxidase-like (*PO*) [69], and *PLA2G15* [70], were significantly induced, indicating that the intestinal antioxidant oxidation functions were dysregulated in *F. chinensis*. *PLA2G15*, a lysosomal specific phospholipase A2, was purified from a bovine brain and plays a primary role in host defense [70]. Cathepsins, the predominant lysosomal proteases, play a vital role in physiological processes and several diseases [71,72]. For example, the upregulation of *CTSB* and *pCTS-L*, which have antioxidant properties, was reinforced in the shrimp intestine to counteract inflammatory disorders. *CD109* and *TNFAIP8* were also involved in the anti-pathogenesis of inflammatory diseases [73,74]. *PT-1* [75] can protect the host from pathogen damage, which is beneficial for coping with low-salinity stress. *Casp1* is a gene encoding a variety of proteins associated with cell death, and the inhibition of its expression impairs the proliferation of disease-associated cells [76]. With the down regulation of the apoptosis gene *Casp 1* [77,78], it is reasonable to speculate that hosts with *F. chinensis* are undergoing an inflammatory process and respond to oxidative stress to alleviate some of the effects of low-salinity stress. However, further efforts are required to study the key intestinal immune genes. 

### 4.3. Relationship between Intestinal Microbial Community and Expression of Immune-Related Genes

To further elucidate host–microbe relationships, we performed a correlation analysis between intestinal immune-related genes and the microbiota by calculating the Pearson coefficient. Of these 12 immune genes, the upregulated expressions of *PT-1*, *pCTS-L*, *PLA2G15*, *CTSB*, and *TNFAIP8* were positively correlated with the genera *Lactococcus*, *Lachnospiraceae_NK4A136_group*, *Pseudomonas*, and *Ralstonia*, suggesting that these bacteria might contribute to the intestinal immune homeostasis of *F. chinensis* under low-salinity stress. Overall, the above microbiota genera were significantly positively correlated with the downregulation of *Muc-2*, while they were significantly negatively correlated with the downregulation of *Casp1*. Therefore, the two genes were significantly altered after low-salinity stress, which might be a biomarker for the health status of shrimp.

Taken together, the microbial biomarkers/genes identified in this study could be applied to monitor the health status of *F. chinensis* under acute salinity stress, and screened as tolerance-related markers to better understand the molecular mechanisms of internal immunity in shrimps.

## 5. Conclusions

Our results showed that low-salinity stress could induce positive changes in intestinal microbiota composition and gene expression. Low-salinity stress disrupts intestinal homeostasis in shrimp, and stimulates the expression of genes related to energy metabolism, immune and digestive systems, and glycan biosynthesis. There is a connection between the intestinal microbial composition and the expression of immune-related genes under low-salinity stress. These findings provide new perspectives for ameliorating salinity stress in the shrimp industry.

## Figures and Tables

**Figure 1 biology-12-01502-f001:**
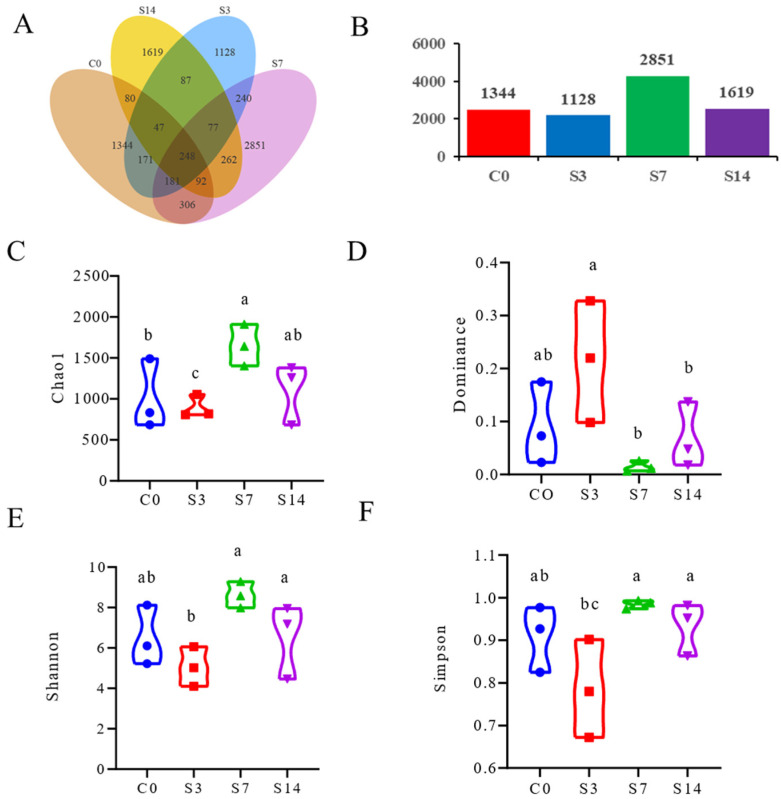
Abundance and diversity of the intestinal microbial community in *F. chinensis* under low-salinity stress. (**A**) The numbers of shared Amplicon Sequence Variants (ASVs) indicated by a Venn diagram. (**B**) The numbers of unique ASVs indicated by a bar chart. (**C**) Chao1 index, (**D**) Dominance index, (**E**) Shannon index, and (**F**) Simpson index. The same lowercase letter means the differences are not significant, and the different lowercase letters mean the differences are significant (*p* < 0.05).

**Figure 2 biology-12-01502-f002:**
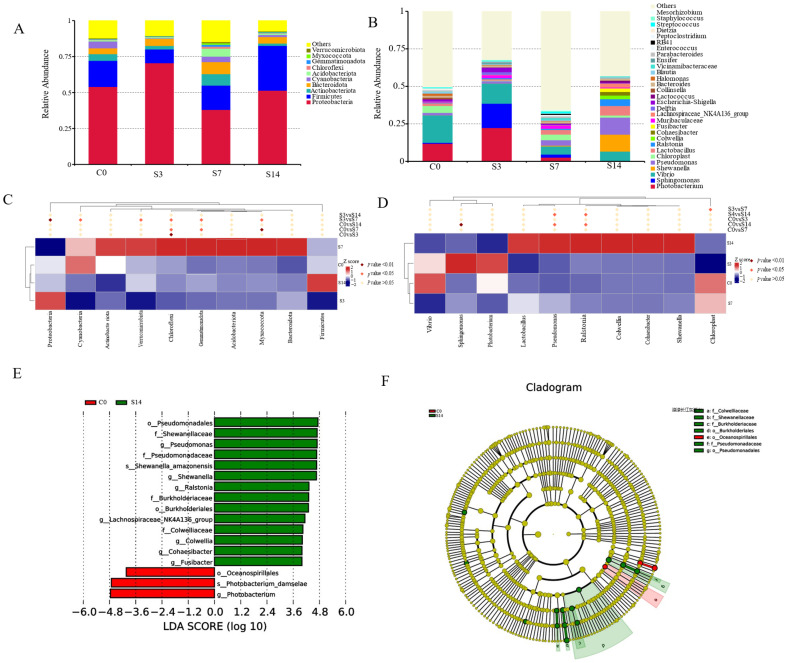
Intestinal microbiota community analysis of *F. chinensis* under low-salinity stress. (**A**) Relative abundances of the top 10 dominant phyla. (**B**) Relative abundances of the top 30 dominant genera. (**C**) Heatmap represents the level of phyla significantly altered. (**D**) The heatmap represents the level of genera significantly altered. (**E**) ASV markers found in the intestinal bacterial community of *F. chinensis* exposed to CO (red bars) and S14 (green bars) group. LDA score (above 4.0) of different bacterial taxa. (**F**) LefSe cladogram plot.

**Figure 3 biology-12-01502-f003:**
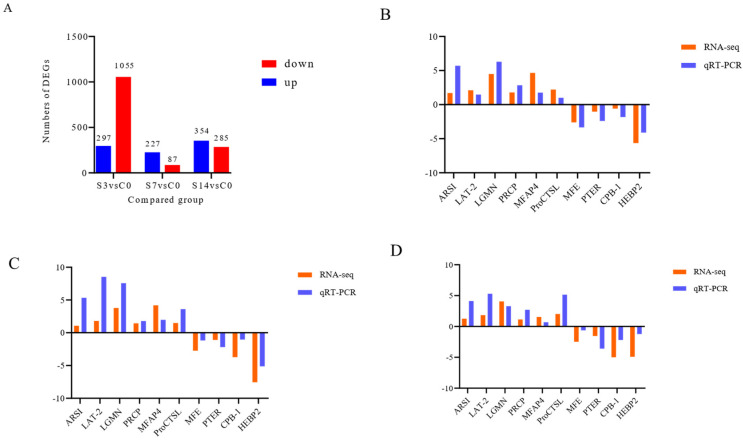
Verification of the differentially expressed genes (DEGs) identified by the intestine transcriptome analysis of *F. chinensis* under low-salinity stress. (**A**) The number of DEGs identified in the three samples compared to the control group (C0). (**B**) Expression levels of the 10 DEGs were verified in S3 vs. C0 by qRT-PCR. (**C**) The expression levels of the 10 DEGs were verified in S7 vs. C0 by qRT-PCR. (**D**) The expression levels of the 10 DEGs were verified in S14 vs. C0 by qRT-PCR.

**Figure 4 biology-12-01502-f004:**
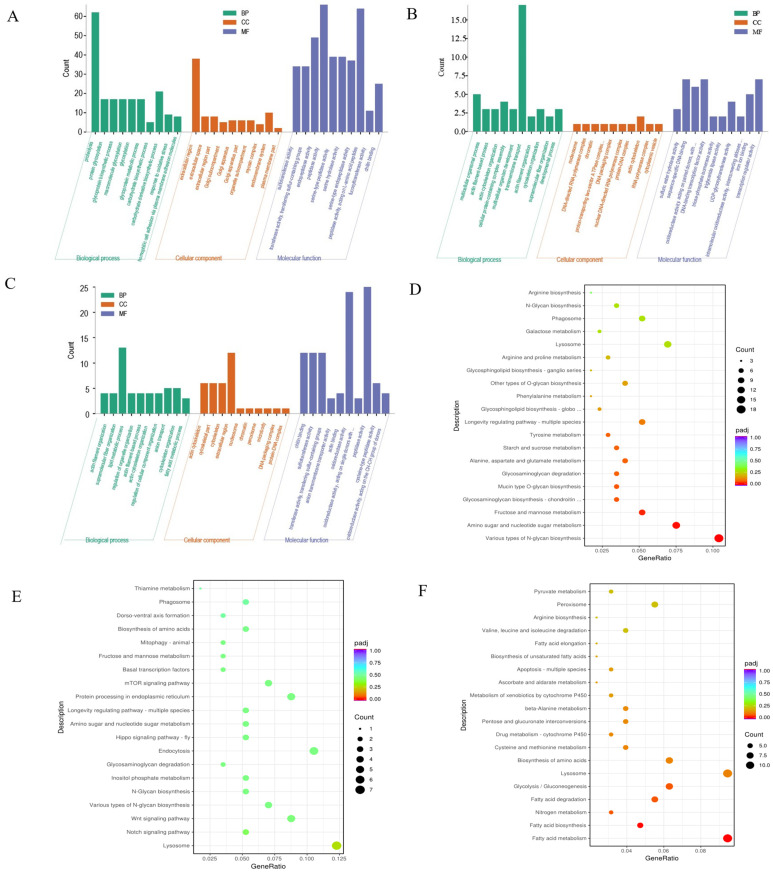
Enrichment analysis of differentially expressed genes (DEGs). (**A**) Enrichment results of Gene Ontology (GO) terms in S3 vs. C0. (**B**) Enrichment results of GO terms in S7 vs. C0. (**C**) Enrichment results of GO terms in S14 vs. C0. Red represents biological processes (BP); green represents cellular components (CC); and blue represents molecular functions (MF). (**D**) The top 20 KEGG pathway analysis results in S3 vs. C0. (**E**) The top 20 Kyoto Encyclopedia of Genes and Genomes KEGG pathways analysis results in S7 vs. C0. (**F**) The top 20 KEGG pathways analysis results in S14 vs. C0.

**Figure 5 biology-12-01502-f005:**
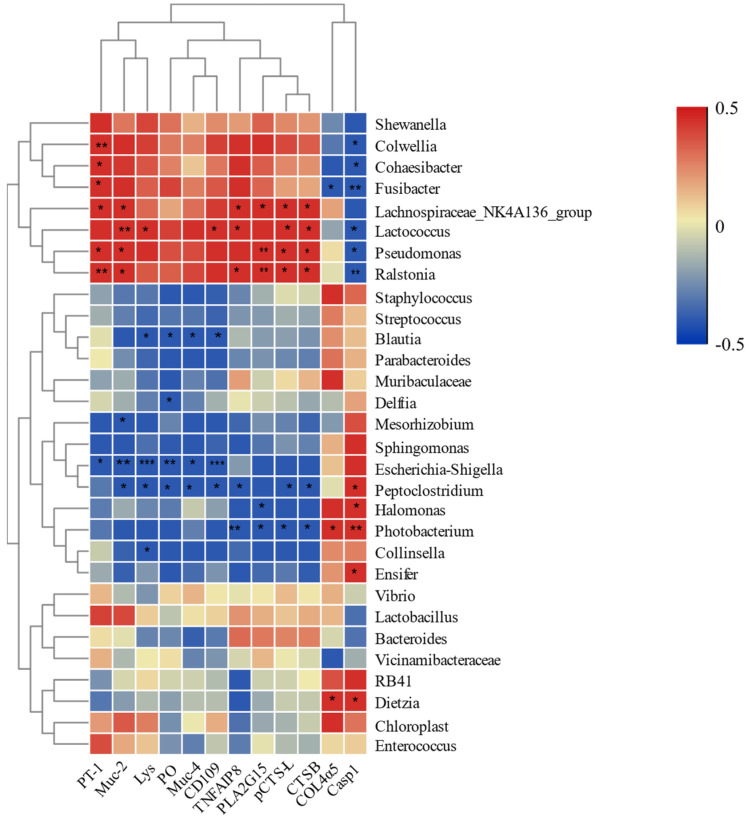
Significant correlation between intestine bacteria at the genus level and immune-related DEGs. Red: positive correlation; blue: negative correlation (* *p* < 0.05; ** *p* < 0.01; *** *p* < 0.001).

## Data Availability

The 16S rDNA sequencing and transcriptome sequencing data in this study will be made available on request.

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
