# Peer review of "Integrated Analysis of the Intestinal Microbiota and Transcriptome of Fenneropenaeus chinensis Response to Low-Salinity Stress"

_biology, 2023, doi:10.3390/biology12121502_

Round 1
Reviewer 1 Report
Comments and Suggestions for Authors
This work is about study of interaction between gut microbiome and transcriptome in F. chinensis after getting responded to low salinity. This study found alteration of relative abundance of set of bacteria and transcriptomic changes under low salinity stress. These omics interaction were analzyed, and it provided interesting results. However, there are some points that need to be corrected and given more explaination.
1) In section 2.1, How many shrimp culture in each group and how many shrimp used to collect samples in each group?
2) In the section 2.1, how many shrimp were cultured in each group before selecting only 3 shrimp for omics? Were they cultured just only 3 shrimp in each group?
3) Why S3 had lower alpha diversity while S7 and S14 increased?
4) In section 3.3 (lines 238-239), why were these 12 DEGs chosen to compare with microbiota? They were different set of gene when compared with the validation of DEG in Figure 3.
5) In the section 2.5, author mentioned using correlation analysis between DEGs and bacterial data. What criteria or approach to generate interaction result between DEGs and microbiota? Were Up- or Down-regulation of DEGs together with high or low abundance of bacteria? please give more information about this analysis. Also, author also mentioned filter of DEGs and used p-value <0.01 but in line 143, p-value was set at 0.05. What were differences between these analyses? In addition, line 138-139 were about analysis of correlation between ASV and DEG but there were DEG analysis in the later lines. They like different or not related detial. Please explain or describe this clearer.
6) Please give more discussion and some literatures about example of bacteria (line 337) negatively correlated to candidate genes (Mu-2 and/or Mu-4, casp1), and Why and how those potential probiotics negatively correlated to casp1?
Minor points
1) Please provide a statement of ethic of animal use in the section 2.1.
2) Line 151, 154, 172, 180: Please correct "Figure" in entire maniscript.
3) In figure 1A, it is diffucult to see the detail in this figure. The label is too small and not clear even increase of magnificance. Please correct this one.
4) About resolution of the figures, I'm not sure that resolution of figures in the manuscript are compressed to make this file or not. I cannot see some figure clearly.
5) Please carefully check in the reference section, all bibliography have "[J]" at the end of title. Are those normal for the journal format? Also, please correct reference No. 34 from ";" to ",".
Reviewer 2 Report
Comments and Suggestions for Authors
The manuscript reports the integrated analysis of intestinal microbiome and intestinal gene expression profiles from Fenneropenaeus chinensis during low-salinity stress. The comments and suggestions are as follows:
1. The primer table was not mentioned in the method
2. FigureS1, the line/dot color were not labeled and explained.
3. Please more clearly explain the result in Fig.1. Shannon and Simpson were not shown the significant different between S3 and C0. Moreover, S7 and S14 were not mentioned.
4. According to NGS and qRT-PCR in Fig.3, the error bars were not presented. How many replicated experiments did you perform?
5. Author did not mention about gene names that selected for qRT-PCR.
6. Why did author analyze the gene expression of immune-related DEGs by qRT-PCR.
7. S7 need to be discuss in 4.1 session.
8. As an important immune organ, the intestine maintains homeostasis under conditions of low-salinity stress by regulating the expression of immune-related genes. Please reference?
9. The genes that author discussed in 4.2 were not appeared in transcriptome and qPCR results.
10. Only 7 genes from 12 immune-related genes were discussed in 4.3
Reviewer 3 Report
Comments and Suggestions for Authors
Comments on the manuscript with ID (biology-2656527).
Q1. Abstract: A brief overview on the used methodology and exposure salinities should be described in the abstract section instead of lines 27-28.
Q2. Lines 41-43: I disagree with your conclusion. As expected, any change in the rearing environment of shrimp will lead to a change in the intestinal microbiome. Therefore, the authors should define what we will get or occur from this change. You should define the best rearing salinity. What did the obtained changes in the intestinal microbiome result in? This information will lead to another conclusion.
Q3. Lines 72-73: “For fish, some intestinal microbiota can be utilized as biomarkers to predict health impacts caused by the variations in environmental elements”. These lines should be deleted as not related to the subject of shrimp response to salinity or external stressors.
Q4. Lines 77-78: “A white spot syndrome virus (WSSV) challenge in F. chinensis revealed several immune-relevant genes [31-33]. These lines and associated references should also be deleted. You did not examine the response of Chinese shrimp to WSSV challenge. This information is also away from the subject of your study.
Q5. Line 88: The water quality parameters should be presented as means ± SE. Number of samples must be described.
Q6. Lines 91-92: Add unit for measuring salinity.
Q7. Add a suitable reference for the salinity used in the control group.
Q8. How the authors acclimate the shrimp to be exposed with 15 ppt salinity. How you decrease from the control 30 ppt to 15 ppt without affecting the health status and welfare of shrimp? These details must be clearly defined for replication of your findings in the future by other researchers.
Q9. Line 97: Add detail on how the authors stored the extracted intestinal tissues.
Q10. Line 135: Add a suitable reference related to the methodology (the 2-ΔΔCt method)?
Q11. Did you find the β-actin as a single housekeeping gene is sufficient instead of using two housekeeping genes?
Q12. Line 151, Line 154, Line 172, Line 180, Line 185, Line 188, Line 198, Line 203, and in other lines: Revise “Error! Reference source not found”.
Q13. Comments on reference section
There are several comments. You should revise every single reference carefully.
ü Latin names should be written in italics.
ü Revise the Journal names.
ü Delete [J] from the end of all references.
ü You should either write the full name of the journals or abbreviate them.
ü Add the page range in all references.

Extensive English Editing and Proof reading are required
Reviewer 4 Report
Comments and Suggestions for Authors
Review for the paper “Integrated analysis of the intestinal microbiota and transcriptome of Fenneropenaeus chinensis response to low-salinity stress” by Cai Tian, Qiong Wang, Jia Wang, Jie Li, Chen Guan, Yu He, Huan Gao submitted to "Biology".
General comment.
Penaeus chinensis is a significant economic species in northern China, playing a crucial role in both marine fishing and aquaculture. Extensive overfishing has caused a rapid decline in Penaeus chinensis resources, with current annual landings reducing from 40,000 tons to about 3,000–5,000 tons. The current biomass of Penaeus chinensis in the Yellow and Bohai Seas is largely dependent on artificial propagation and release. More than 95% of the fall harvest is from released shrimp. There is a need for new insights into the intestinal microbiota of Penaeus chinensis regarding various factors for successful aquaculture development. The authors conducted a laboratory study to determine the composition and diversity of the intestinal microbiota and transcriptome, focusing on the effects of low salinity on gene expression levels. The authors conducted a laboratory study to reveal the composition and diversity of the intestinal microbiota and transcriptome while focusing on the effects of low salinity on gene expression levels. They attempted to relate these fluctuations with the intestinal microbiota. The authors discovered that low salinity stress can bring about certain favorable adjustments in the composition of intestinal microbiota and gene expression. Salt stress perturbs gut homeostasis in shrimp, activating the expression of genes related to immune and digestion, glycan biosynthesis, and energy metabolism. The authors also detected connections between the immune-related gene expression and the composition of intestinal microbiota in conditions of low salinity stress. These results offer a foundation for further research and could impact significantly on the shrimp industry. The manuscript is composed competently, yet some figures require adjustments. The researchers employed suitable statistical techniques and dissected the key outcomes. With minor modifications, this article is suitable for publication in this journal.
Major concerns.
1) The authors erroneously refer to the species as Fenneropenaeus chinensis. According to the World Register of Marine Species (https://www.marinespecies.org/aphia.php?p=taxdetails&id=246390), the correct name is Penaeus chinensis.
2) Additionally, the authors should provide further details about the shrimp used in this study, such as their sex and maturity status.
3) Figure 1A, Figure 2, and Figure S2 require redrawing because their font size is substantially reduced.
4) The authors referred to PCoA and LDA in the Results section without providing a corresponding explanation of these methods. Therefore, the text must be revised to include this information.
5) In regard to the ANOVA and correlation analysis, it is unclear if the authors tested for normality and homogeneity of variance prior to utilizing these parametric approaches. It is necessary to check these assumptions before using these methods.
Specific remarks.
L 23. Consider replacing “to adapt stress” with “to adapt to stress”
L 61. Consider replacing “can be adapt to higher salinity, and lower-salinity” with “can adapt to higher salinity, and lower salinity”
L 94. Consider replacing “using with light brine” with “using light brine”
L 130. Consider replacing “analyses of the some selected genes” with “analyses of some selected genes”
L 316. Consider replacing “on day14 to alleviate of the salinity-induced intestinal barrier damage, which indicating” with “on day 14 to alleviate of the salinity-induced intestinal barrier damage, which indicates”
L 328. Consider replacing “that host with” with “that hosts with”
L 329. Consider replacing “to alleviates some” with “to alleviate some”
L 330. Consider replacing “to fuctionally study” with “to study”
Comments on the Quality of English LanguageMinor revisions.
Round 2
Reviewer 1 Report
Comments and Suggestions for Authors
All revised points in the revised version are fine for me. I have no further comments.
Author Response
Thank you for your careful comments again.
Reviewer 2 Report
Comments and Suggestions for Authors
This work is suitable to be published in this Journal.
Author Response
Thank you for your suggestions and comments again.
Reviewer 3 Report
Comments and Suggestions for Authors
Comments on the manuscript with ID (biology-2656527-peer-review-v2)
The authors responded well to the comments raised by the anonymous reviewer; however, there are still some points and comments that should be revised. The authors should revise them before the manuscript is considered for publication in Biology.
Q1. Lines 59-60: Add a suitable reference to this sentence.
Q2. Lines 61-62: Delete these lines “However, shrimps can adapt to higher salinity, and lower salinity may cause higher mortality”. You did not provide a reference related to this sentence.
Q3. Line 86: Add methodology and instruments used for measuring salinity, pH, dissolved oxygen, and temperature.
Q4. Line 138: Incorrect citation. Please use this reference (Livak and Schmittgen, 2001).
Livak, K. J., & Schmittgen, T. D. (2001). Analysis of relative gene expression data using real-time quantitative PCR and the 2− ΔΔCT method. Methods, 25(4), 402-408.
Please revise the citations in text and related references. They should be related to the text.
References (minor comments).
Line 408: Haliotis discus – write italic
Line 420: Environmental Science and Pollution Research International
Line 426: Journal of Trace Elements in Medicine and Biology
Line 430: Journal of Comparative Physiology. B, Biochemical, Systemic, and Environmental Physiology
Line 442: The Science of the Total Environment
Line 448: BMC Microbiology
Line 453: Nature Metabolism
Lines 461-462: Journal of Comparative Physiology. B, Biochemical, Systemic, and Environmental Physiology
Line 529: The Journal of Biological Chemistry
Line 539: pH

Extensive editing of English language required
